# Gentrification and Air Quality in a Large Urban County in the United States

**DOI:** 10.3390/ijerph20064762

**Published:** 2023-03-08

**Authors:** Hollis Hutchings, Qiong Zhang, Sue Grady, Lainie Mabe, Ikenna C. Okereke

**Affiliations:** 1Department of Surgery, Henry Ford Health System, 2799 W. Grand Blvd, Detroit, MI 48202, USA; 2Department of Public Health Sciences, Henry Ford Health, Detroit, MI 48202, USA; 3Department of Geography, Environment and Spatial Sciences, Michigan State University, East Lansing, MI 48824, USA; 4School of Medicine, University of Texas Medical Branch, Galveston, TX 77555, USA

**Keywords:** air pollution, gentrification, monitoring, disparities

## Abstract

**Introduction:** Increases in industrialization and manufacturing have led to worsening pollution in some components of air quality. In addition, gentrification is occurring in large cities throughout the world. As these socioeconomic and demographic changes occur, there have been no studies examining the association of gentrification with air quality. To investigate this association, we studied the trends of gentrification, changes in racial distribution and changes in air quality in each zip code of a large urban county over a 40-year period. **Methods:** We conducted a retrospective longitudinal study over 40 years in Wayne County, Michigan using socioeconomic and demographic data from the National Historical Geographic Information System (NHGIS) and air quality data from the United States Environmental Protection Agency (EPA). To assess gentrification, longitudinal analyses were performed to examine median household income, percentage with a college education, median housing value, median gross rent and employment level. The racial distribution was evaluated in each zip code during the time period. Gentrification was studied in relation to air quality using nonparametric 2-sample Wilcon–Mann–Whitney tests and Binomial Generalized Linear Regression models. **Results:** Although air quality improved overall over the 40-year period, there was a lesser rate of improvement in gentrified areas. Furthermore, gentrification was strongly associated with racial distribution. The most substantial gentrification occurred from 2010 to 2020, in which a specific cluster of adjacent zip codes in downtown Detroit experienced intense gentrification and a drop in the percentage of African-American residents. **Conclusions:** Gentrified areas seem to have a less pronounced improvement in air quality over time. This reduction in air quality improvement is likely associated with demolitions and the construction of new buildings, such as sporting arenas and accompanying traffic density. Gentrification is also strongly associated with an increase in non-minority residents in an area. Although previous definitions of gentrification in the literature have not included racial distribution, we suggest that future definitions should include this metric given the strong association. Minority residents who are displaced as a result of gentrification do not experience the improvements in housing quality, accessibility to healthy foods and other associations of gentrification.

## 1. Introduction

The rise in industrialization, traffic density and population density has led to worsening air quality over the last 100 years. Previous studies have shown that these changes are linked to medical illnesses, such as asthma and lung cancer [1,2,3,4]. Many of these studies have also shown that there are significant disparities in incidence and outcome with these disease processes [5,6]. Minorities have tended to have higher mortality rates from illnesses, such as asthma and lung cancer [7,8].

Gentrification is the process by which neighborhoods in traditionally low-income areas experience a surge in economic investment and an inflow of high-income people [9]. These changes in resident distribution are often accompanied by better housing options, improved access to healthy foods, improved roads and transportation systems and multiple other improvements in quality of life [10]. Although many of these benefits have been documented to occur with gentrification, the changes in air quality with gentrification have not previously been studied.

Importantly, it appears that structural improvements to the community occur prior to the initial phases of gentrification. One interpretation of the “Environmental risk transition theory” posits that investments in the community lead to the migration of suburban residents into urban areas [11]. This influx of new residents spurs the continued investment into industries, such as restaurants and stores. Given the negative associations of air quality with disease, our goal was to analyze the relationships of gentrification with air quality over a 40-year period in a large urban county in the United States. 

The city of Detroit, Michigan is the largest city in Wayne County. It has experienced dramatic changes over the last 70 years. The population of the city declined from 1.3 million people in 1950 to 740,000 people in the last United States census tabulation in 2020 [12]. The automotive industry and other manufacturing plants have changed locations during this period, which has affected air quality. Following the economic crisis in the early part of this century, the city of Detroit declared bankruptcy in 2013. Soon after, however, there was an increase in economic investment in the downtown area and a migration of young professionals to this area. Our goals were to describe the trends in gentrification in Detroit/Wayne County, including changes in the racial composition and to examine any associations between gentrification and air quality over time.

## 2. Methods

### 2.1. Gentrification

After institutional review board approval was obtained (HFHS IRB-15286), the National Historical Geographic Information System (NHGIS) was queried for population-level sociodemographic data to examine underlying populations from 1980 to the present. Information was obtained and evaluated for each zip code in Wayne County (n = 68) during the time period [13]. The year 1980 was the first year that data were available. To assess gentrification, we adopted a longitudinal approach first presented by the Urban Health Collaborative at Drexel University and adopted for the Wayne County study area [14]. This definition did not include race as one of the criteria to label gentrification. First, the eligibility to gentrify was determined by identifying those zip codes with a median household income less than the 75th percentile within Detroit (n = 27) and outside of Detroit (n = 41) in 1980 (base). Zip codes were stratified within and outside of Detroit due to significant differences in the median household income distributions and the need to study gentrification within each zip code relative to its geography in Wayne County. Population and housing characteristics were studied for those eligible zip codes, including (a) the percentage of persons with a college education (4 years or more) or a professional degree, (b) median housing value and (c) median gross rent. If the percentage of college-educated/professionals AND median housing value OR median gross rent fell within the 50-75th percentile of their distributions, then the zip code was labeled as gentrified. If the percentage of college-educated/professionals AND median housing value OR median gross rent fell above the 75th percentile of their distributions, then the zip code was labeled as intensely gentrified. Median household income, median housing value and median gross rent for 1980, 1990, 2000 and 2010 were adjusted for inflation based on June 2020 inflation estimates (United States Bureau of Labor Statistics, 2022) in order to compare these characteristics across the decades. The reference categories for each of these characteristics are provided in Table 1. To assess spatial and temporal changes in the non-Hispanic African American or Black population in relation to other changes, a crosshatch was used to show those zip codes with >70 percent non-Hispanic Black population at the base time period (black crosshatch) and subsequent time period (red crosshatch). State roads in Wayne County were also displayed to show the relationship between the county’s road network and gentrification. 

### 2.2. Air Quality

Daily air quality-pollution data from 1980 to 2021 were obtained from the United States Environmental Protection Agency (EPA) AirData website for Michigan. The data were retrieved from the Air Quality System database created to assist health and policy research [15]. Since only historical and publicly available air quality data were collected, results were limited by the number of sensors present at the time of data acquisition. As time progressed, the density of air sensors placed by EPA increased. A total of 10 pollutant levels were collected, including carbon monoxide, ozone, particulate matter 2.5 (PM_2.5_), particulate matter 10 (PM_10_), nitrogen dioxide (NO2), nitrous oxide, sulfur dioxide (SO2), lead, volatile organic compounds (VOCs) and hazardous air pollutants (HAPs). All pollution data were geocoded based on the location of EPA air monitors followed by empirical Bayesian Kriging modeling to estimate and create a continuous surface of pollution density for each air pollutant [16]. The average value was calculated and assigned back to each zip code area by using zoning functions. The spatial geocoding, estimation, calculation and assignment were completed in ArcGIS Pro (ESRI, ArcGIS Pro 2.8.3, Redlands, CA).

### 2.3. Statistical Analysis

The population and housing variables to define gentrification were not normally distributed and zip codes were independent of each other. As such, nonparametric 2-sample Wilcoxon–Mann–Whitney tests were used to evaluate decadal differences in population characteristics between zip codes within Detroit and outside Detroit. These analyses are described further in the footnote to Table 1. A similar approach was used to test the decadal differences in variation for each pollutant. Furthermore, the Kruskal–Wallis H tests were used to examine the difference among the 10 air pollutants. Analyses were performed for 40 years in all areas, followed by subgroup analyses, including 40 years in zip codes with (a) gentrification, (b) intense gentrification and (c) zip codes with no gentrification. Binomial General Linear Models (GLMs) were utilized to determine the direction, magnitude and significance of the gentrification and air pollution relationships over time. The results are described further in the footnote to Table 2. Wilcoxon testing was used to compare the changes in air quality in Gentrification Alley versus areas outside Gentrification Alley.

## 3. Results 

### 3.1. Population Characteristics

Table 1 lists the overall population and housing characteristics. The population in the city of Detroit dropped 43 percent, from 1.8 million people in 1980 to 700,000 people in 2020. The percentage of Detroit residents who were African American rose from 61 percent in 1980 to 78 percent in 2010, and then dropped to 72 percent in 2020. The Hispanic population increased from 3 percent in 1980 to 8 percent in 2020, while the Asian population remained relatively constant. Between 2010 and 2020, the percent of college-educated adults rose within the city of Detroit, while the level decreased slightly outside Detroit. Median housing value was substantially lower and gross rent was substantially higher in Detroit compared to outside of Detroit in Wayne County across the study time period. 

### 3.2. Gentrification

Figure 1 details levels of gentrification during each decade in each zip code of Wayne County with an overlying state road map. In 1990, intense gentrification occurred in Wayne County’s (suburban) zip codes west of the city of Detroit. This intense gentrification was maintained through 2020. In the city of Detroit in 1990 and 2010, 2 zip codes experienced intermittent intense gentrification that were near the city’s commercial center. Significant investment into the downtown area, including the construction of three separate large sporting arenas and a major tax reduction for new homes, occurred during this time. 

Thereafter, from 2010 to 2020, 4 zip codes emerged in one cluster of intense gentrification that extended from this city center to include downtown Detroit embedded between two large roadways labeled as “Gentrification Alley” (Figure 2). 

### 3.3. Racial Distribution 

The association between changes in racial distribution and gentrification was analyzed. Specifically, zip codes that experienced intense gentrification also had a substantial displacement of the African-American population between 2010 and 2020 (range—11% to 33% decrease). In fact, each of the 4 zip codes comprising Gentrification Alley were the only zip codes within all of Detroit to have African-American percentages less than 70 percent.

### 3.4. Air Quality vs. Gentrification

To describe variations in air pollution, changes in air pollutant levels were compared in zip codes that experienced gentrification, intense gentrification and no gentrification. The median levels of all air pollutants in all categories demonstrated tremendous decreases over the study period, indicating overall air quality improvement in all areas of Wayne County over the 40-year period. Figure 3 shows an example of the overall trend across all areas of VOCs and SO2. Complete pollutant levels for each year are found in Appendix A. 

To examine the degree of variation, decadal changes were compared between zip codes that experienced intense gentrification, gentrification and no gentrification. Table 2 shows the median of the decadal change in ten pollutants for all zip codes that experienced a certain level of gentrification in Wayne County. Between 2000 and 2010, levels of all pollutants except for tropospheric ozone and HAP decreased in both intense gentrification and gentrification areas of Wayne County. Additionally, between 2010 and 2020, levels of all pollutants except VOCs, ozone and NO_2_ decreased in both intense gentrification and gentrification areas of Wayne County. The Kruskal–Wallis H tests were used to test (1) whether the degrees of variation were the same for all pollutants and (2) whether the total variation of all pollutants were similar in all years. The results showed that in both intense gentrification and gentrification areas, the average changes in air pollutants were significantly different from each other. There was also a statistically significant difference in the average change in all pollutant levels except for tropospheric ozone between 1980 and1990 (*p* < 0.01), 1990 and 2000 (*p* < 0.01), 2000 and 2010 (*p* < 0.01) and 2010 and2020 (*p* < 0.01). 

Wilcoxon testing was used to compare the changes in air quality in Gentrification Alley versus areas outside Gentrification Alley. From 2010 to 2020, there was no significant difference in the change of air pollutant levels in Gentrification Alley compared to outside Gentrification Alley (*p* = 0.99). However, this analysis was restricted by the limited number of years and the number of zip codes in Gentrification Alley.

Next, binomial regression analyses were performed to examine the association between the probability of an area being gentrified and decadal changes in air pollutants. We recorded all zip codes in the county as either gentrified (=1) or non-gentrified (=0). The independent variables were the degrees of variation in air pollutants between 1980 and 1990, 1990 and 2000, 2000 and 2010 and 2010 and 2020. These analyses were performed while controlling for the change in total population, change in the percentage of the African-American population and change in the median income level of each area. When comparing these areas, non-gentrified areas generally had larger improvements in air quality than gentrified areas. Six pollutants (VOCs, PM_2.5_, NO_2_, lead, CO and NO_x_) in particular had a larger mean improvement in non-gentrified areas. Figure 4 shows violin plots of variation of air pollutant variation over time. When examining PM2.5, for example, nearly half of the non-gentrified areas had a large decrease in PM2.5 (range 4–8 units). In contrast, most gentrified areas had only a small improvement (range 0–4 units).

On the *x*-axis, “0” is non-gentrified and “1” is gentrified. VOCs are measured in parts per million. SO_2_ is measured in parts per million. PM_2.5_ and PM_10_ are measured in micrograms per cubic meter. Ozone is measured in parts per billion. NO_2_ is measured in parts per billion. Lead is measured in nanograms per cubic meter. CO is measured in parts per million. HAP are measured in parts per million. NOx are measured in parts per billion.

## 4. Discussion 

Our study showed that gentrification occurred in selected areas of the city and county. Although gentrification was ongoing in Wayne County’s western suburbs of Detroit during all decades in our study, the most marked changes occurred in downtown Detroit between 2010 and 2020. The most likely reasons for gentrification in this area are multi-fold. During this decade, there was substantial economic investment by local and national investors into the city of Detroit [17,18,19]. The influx of capital into these zip codes created hundreds of new supermarkets, restaurants and other resources for residents. However, there was likely a bi-directional effect. It is likely that the change to a wealthier population led to continued investments in entities, such as restaurants and green spaces. Secondly, there was the construction of both a 20,000-seat sporting arena and a 50,000-seat sporting stadium in Gentrification Alley. The construction of these sites led to thousands of new jobs. Finally, there were “tax-exemption zones” created in these zip codes as incentives for people to move to the downtown areas. These tax exemptions were created prior to the intense gentrification seen in Gentrification Alley. As a result of these factors, there was a simultaneous migration into the downtown area of non-minorities with higher education levels and higher incomes displacing African Americans to other areas.

The structure of the roadmap also affected gentrification in Wayne County. The Home Owners’ Loan Corporation (HOLC) was created in 1933 by President Roosevelt to improve the housing market following the Great Depression [20]. The HOLC evaluated hundreds of cities across the United States and created zones based on the perceived risk of lending to residents in each zone. Zones with an “A” grade were considered the best, while zones with a “D” grade were considered the riskiest. Zones with a “D” grade were outlined in red and consisted of mostly minority residents. This zoning led to the term “redlining.” Redlining was historically accompanied by the planning of the interstate highway system [21,22,23]. Through the Federal Aid Highway Act of 1956, interstates were purposefully directed in the middle of the mostly minority “D” zones, thereby creating racially segregated neighborhoods. 

However, this roadmap appears to have facilitated gentrification in Wayne County in recent years. The stadiums and aforementioned sporting arena built early in the gentrification process were built in these specific locations because they were accessible to interstates and large highways [24]. Restaurants and hotels have targeted these locations because consumers have easy accessibility. In addition, prospective residents who have moved from the suburbs to these gentrified areas have appreciated the convenience of being able to drive to extended family in the suburban regions with quick access to the large roadways.

We chose to focus on our home county alone to perform an in-depth analysis of potential trends, which may be related to unique characteristics and nuances of Detroit and Wayne County. This initial analysis served as a pilot for more comprehensive studies. Although we were able to see some patterns related to specific details about Detroit (e.g., gentrification centering around the creation of sporting venues), there will be utility in performing this study on a wider level.

Gentrification has resulted in the displacement of residents from these areas to surrounding zip codes [25,26,27]. These areas have experienced lower levels or limited gentrification resulting in a continued lack of availability of quality housing and healthy food options. These areas also have not been granted tax exemptions as seen in gentrified areas. Future efforts should focus on improving access to resources in these areas with displaced people from gentrified areas. In addition, tax incentives should be considered in these marginalized areas.

Our initial hypothesis was that gentrification would be associated with improved air quality. We expected gentrification to be associated with increased parks and greenspaces. However, our study showed the opposite phenomenon. It should be stressed that although there may be an association between gentrification and air quality, there are multiple factors that contribute to air quality. It is inappropriate to claim that gentrification or change in racial distribution alone accounts for the changes in pollutant levels. Nevertheless, it is useful to understand and be able to predict potential changes that may occur in areas that are simultaneously undergoing gentrification. 

Although air quality improved over the course of 40 years throughout the county, the improvements were less substantial in intensely gentrified and gentrified areas. In retrospect, this trend fits with many of the proposed causes of gentrification. The demolition of older buildings and construction of a number of businesses and sporting facilities has likely led to added dust and increased traffic density in gentrified areas. The increased usage of electricity and gas in these areas has likely contributed to increased pollutant levels. It is true that current pollutant levels are improved overall compared to 1980 [28,29,30,31]. Our study also showed that gentrified areas are not improving pollution levels as rapidly as non-gentrified areas.

There are several quantifiable points of evidence that may support the trend of gentrified areas not improving air quality as rapidly. In the United States, domestic demand for gasoline rose from 110 billion gallons to almost 150 billion gallons per year between 1990 and 2020 [32]. Much of this increase was secondary to increased traffic density in urban areas. In Detroit, the roadway congestion index rose from 0.91 in 1982 to 1.09 in 2011 [33]. In a study that evaluated 83 different urban areas, vehicle-miles traveled was projected to increase 33 percent from 2000 to 2030 [34].

Our study showed a very strong relationship between gentrification and racial distribution. The origin of the term gentrification can be traced back as far as the ancient Roman and Northern African regions, during which large upper-class country houses were converted into farming compounds but ultimately back to more sophisticated upper-class country homes [35]. Most official definitions of gentrification have not included race. However, in our study, race was shown to be strongly associated with gentrification. Given the disparities in access to affordable housing, nutritious food and other social determinants of health, we advocate that future definitions of gentrification should include racial distribution as a criterion.

This study raises multiple questions for future investigation. As poor air quality is associated with multiple illnesses [36,37,38,39], how have recent trends in gentrification impacted the change in the incidence of diseases in these areas? Moreover, does the increased wealth and accessibility to gentrified communities mitigate or attenuate some of the effects of poor air quality? In addition, how do displaced residents from gentrified communities suffer as a result of the migration and what is its contribution to racial health disparities?

There are limitations to our study. Gentrification was measured over the last 40 years at the end of each decade, during which time the sources of census data have changed from the decadal census (SF1 1980, 1990, 2000) to surveyed census data (ACS 5-year retrospective estimates at 2010 and 2020), so some decadal changes in gentrification and racial composition may be due to sampling bias. This study was based on air quality sensors from the EPA, but the overall density of sensors is fairly low. As air quality research has evolved over the last decade, the development of private air quality networks has increased. With the implementation of more sensors, a finer air quality network can be established for better data collection and thus stronger analyses. Despite the limitation of a historic low density of EPA-established air quality sensors, this study demonstrates why further work is needed to develop a finer air quality network for further study into the effects of air quality and disease processes. Evaluation of the changes in Gentrification Alley compared to areas outside this region was limited by having only 10 years and a limited number of sensors to compare changes. More air monitoring sensors are needed to have a better understanding of air quality at a neighborhood level to thereby improve future air pollution exposure and health outcome assessments. 

## 5. Conclusions

Over the 40-year time period studied, we demonstrated that there were areas in Wayne County that experienced intense gentrification and gentrification, most pronounced in the last 10 years of the study period in the city of Detroit. These areas of Detroit that experienced intense gentrification did not experience as dramatic of an improvement in air quality as non-gentrified areas. In areas of Detroit that experienced intense gentrification, we also observed the displacement of African Americans. Further work is needed that includes racial distribution in the definition of gentrification as well as understanding more fully the impact that air quality has when comparing areas with and without gentrification. This work would be further enhanced by deploying more air sensors throughout these changing communities.

## Figures and Tables

**Figure 1 ijerph-20-04762-f001:**
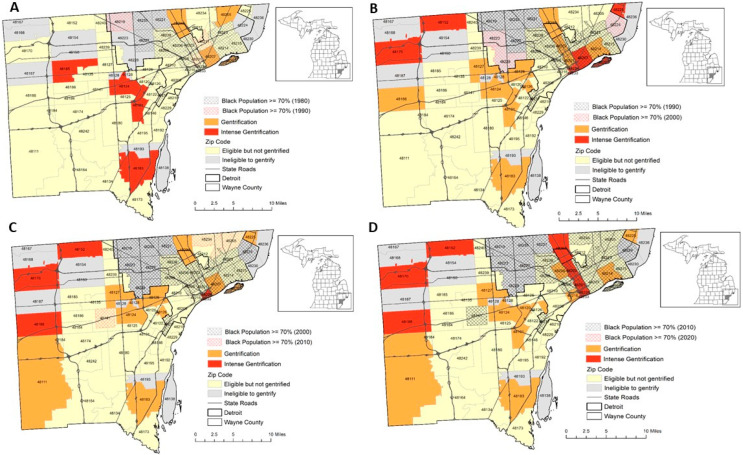
Gentrification in Wayne County from (**A**) 1980 to1990, (**B**) 1990 to2000, (**C**) 2000 to2010 and (**D**) 2010 to2020.

**Figure 2 ijerph-20-04762-f002:**
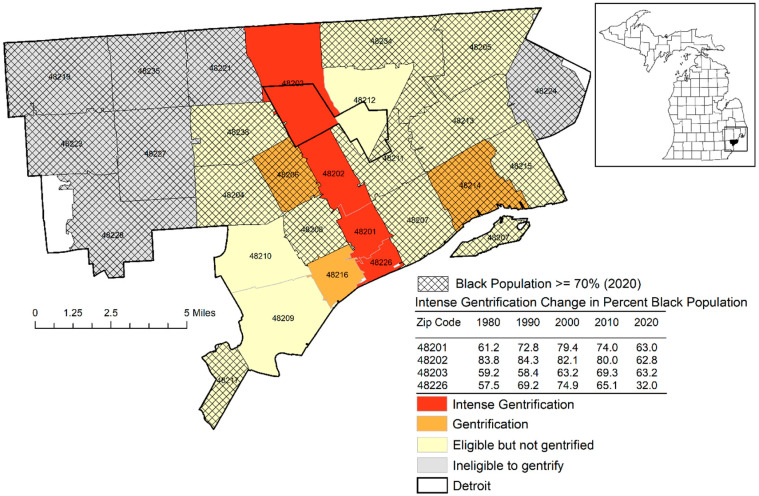
A cluster of 5 contiguous zip codes labeled “Gentrification Alley”, which experienced intense gentrification from 2010 to 2020.

**Figure 3 ijerph-20-04762-f003:**
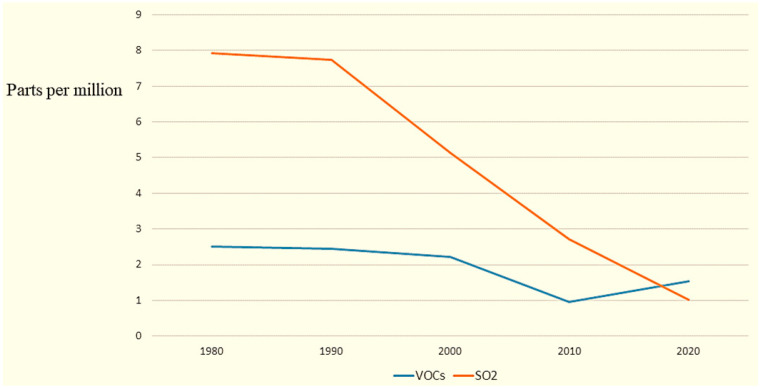
Pollutant levels from 1980 to 2020 for VOCs and SO2.

**Figure 4 ijerph-20-04762-f004:**
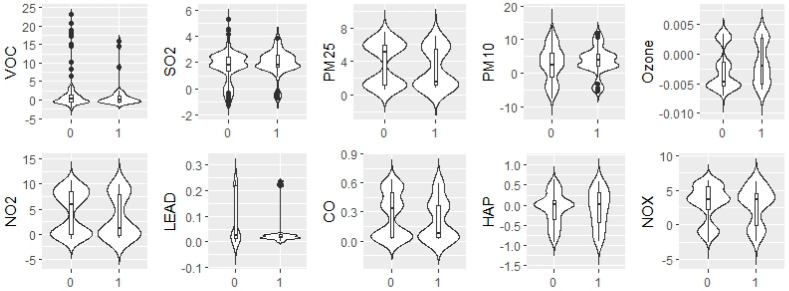
Violin-Box Plot of Air Pollutant Variation.

**Table 1 ijerph-20-04762-t001:** Characteristics of Gentrification by Decade in Wayne County, Michigan from 1980 to 2020.

Gentrification Characteristics	Year				
	1980	1990	2000	2010	2020
Median Household Income ^1^-Top Quartile (Eligibility to Gentrify)					
Within Detroit (≥75th percentile)	$54,590	$46,254	$51,604	$37,295	$32,533
Outside Detroit (≥75th percentile)	$84,720	$87,622	$96,252	$85,989	$72,357
Race & Ethnicity					
Within Detroit					
Black (%)	60.7	72.0	77.3	77.6	72.3
White (%)	34.9	22.9	15.1	13.1	16.2
Asian (%)	1.1	1.3	1.7	2.1	2.8
Other (%)	0.2	0.1	0.2	0.2	0.5
Hispanic (%)	3.0	3.6	5.7	7.1	8.3
Outside Detroit					
Black (%)	4.4	5.8	8.4	15.0	17.1
White (%)	99.1	90.4	86.0	77.0	72.3
Asian (%)	1.1	1.7	2.5	3.2	4.0
Other (%)	0.1	0.1	0.3	0.1	0.4
Hispanic (%)	1.5	2.1	2.9	4.5	6.1
Percent College Educated or Professional Degrees ^2^					
Within Detroit (50–75th percentile)	4.0	5.4	6.3	8.3	11.0
Within Detroit (≥75th percentile)	8.2	9.6	10.6	12.3	18.9
Outside Detroit (50–75th percentile)	7.7	10.0	12.2	14.2	14.0
Outside Detroit (≥75th percentile)	12.8	18.3	24.4	27.9	26.1
Median Home Value ^3^					
Within Detroit (50–75th percentile)	$62,054	$48,874	$81,667	$87,738	$57,027
Within Detroit (≥75th percentile)	$81,114	$65,049	$116,383	$110,845	$83,600
Outside Detroit (50–75th percentile)	$150,052	$138,474	$199,030	$177,227	$118,800
Outside Detroit (≥75th percentile)	$195,275	$184,088	$260,308	$232,374	$215,129
Median Gross Rent ^4^					
Within Detroit (50–75th percentile)	$165	$721	$719	$878	$786
Within Detroit (≥75th percentile)	$275	$859	$820	$982	$906
Outside Detroit (50–75th percentile)	$104	$986	$967	$957	$892
Outside Detroit (≥75th percentile)	$262	$1141	$1183	$1115	$1041
Total Population Change					
Within Detroit (N base, % change)	1,317,850	−13.53	−9.45	−38.54	−43.91
Outside Detroit (N base, % change)	1,148,812	−3.96	−1.52	−1.69	2.46

^1^ Distributions based on Independent-samples Mann–Whitney U Tests were significantly different (*p*-value < 0.001) within Detroit and outside of Detroit across decades. ^2^ Distributions based on Independent-samples Mann–Whitney U Tests were significantly different for 1980 (*p*-value = 0.006), 1990 (*p*-value = 0.004), 2000 (*p*-value < 0.001), 2010 (*p*-value < 0.001) and 2020 (*p*-value = 0.039) within Detroit and outside of Detroit across decades. ^3^ Distributions based on Independent-samples Mann–Whitney U Tests were significantly different (*p*-value < 0.001) within Detroit and outside of Detroit across decades. ^4^ Distributions based on Independent-samples Mann–Whitney U Tests were significantly different for 1990 (*p*-value < 0.001), 2000 (*p*-value < 0.001), 2010 (*p*-value = 0.025) and 2020 (*p*-value = 0.025). In 1980, the distributions were not significantly different (*p*-value = 0.101) within Detroit and outside of Detroit across decades.

**Table 2 ijerph-20-04762-t002:** Median changes of pollutants in different areas of Wayne County during each decade.

Air Pollutants	Year			
	1980–1990 ^12^	1990–2000 ^12^	2000–2010 ^12^	2010–2020 ^12^
Intense Gentrification (Median) ^10^				
CO ^7^	0.576	0.361	0.075	0.029
HAP ^8^	0.019	−0.819	−0.402	0.333
LEAD ^6^	0.234	0.026	0.002	0.016
NO_2_ ^5^	7.415	0.561	9.046	−0.411
NOX ^9^	3.633	−0.298	5.726	3.660
Ozone ^4^	−0.005	−0.002	−0.005	0.003
PM_10_ ^3^	−3.838	5.012	6.059	2.147
PM_2.5_ ^3^	--	--	5.313	1.246
SO_2_ ^2^	−0.523	2.277	2.264	1.687
VOCs ^1^	--	0.208	1.394	−0.571
Gentrification (Median) ^11^				
CO ^7^	0.588	0.359	0.072	0.032
HAP ^8^	0.028	−0.447	−0.298	0.406
LEAD ^6^	0.223	0.027	0.012	0.017
NO_2_ ^5^	6.128	0.939	9.478	−0.622
NO_X_ ^9^	4.439	−1.130	6.028	3.527
Ozone ^4^	−0.006	−0.002	−0.005	0.003
PM_10_ ^3^	−4.623	5.036	9.300	2.081
PM_2.5_ ^3^	--	--	6.022	1.269
SO_2_ ^2^	1.572	2.725	2.508	1.731
VOCs ^1^	--	0.358	1.399	−0.482

^1^ VOCs are measured in parts per million. ^2^ SO_2_ is measured in parts per million. ^3^ PM_2.5_ and PM_10_ are measured in micrograms per cubic meter. ^4^ Ozone is measured in parts per billion. ^5^ NO_2_ is measured in parts per billion. ^6^ Lead is measured in nanograms per cubic meter. ^7^ CO is measured in parts per million. ^8^ HAP are measured in parts per million. ^9^ NOx are measured in parts per billion. ^10^ Kruskal–Wallis H tests proved that the average changes in different types of air pollutants were statistically significant in the Intense Gentrification area (*p*-value < 0.001). ^11^ Kruskal–Wallis H tests proved that the average changes in different types of air pollutants were statistically significant in Gentrification area (*p*-value < 0.001). ^12^ Kruskal–Wallis H tests proved that the average changes in all air pollutants were statistically significant in 1980–1990 (*p*-values < 0.001), 1990–2000 (*p*-values < 0.001), 2000–2010 (*p*-values < 0.001), 2010–2020 (*p*-values < 0.001).

## Data Availability

Population-level sociodemographic data were obtained from the National Historical Geographic Information System (NHGIS). https://www.ipums.org/projects/ipums-nhgis/d050.v15.0 (accessed on 3 December 2022). Air quality data were obtained from the United States Environmental Protection Agency (EPA) AirData website for Michigan. https://www.epa.gov/outdoor-air-quality-data (accessed on 10 March 2022).

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
