# Peer review of "Gentrification and Air Quality in a Large Urban County in the United States"

_ijerph, 2023, doi:10.3390/ijerph20064762_

Round 1

Reviewer 1 Report

This paper studies the impact of gentrification of Wayne County, MI on air quality and racial distribution changes in the past 40 years. The authors applied multiple statistical tests to compare the differences between air quality and racial distribution between intense gentrification, gentrification, and non-gentrification zip code areas. They constructed the binomial general linear models to determine the direction, magnitude, and significance of the gentrification and air quality relations. They concluded that gentrification areas seem to have less pronounced improvement in air quality over time. Gentrification is also strongly associated with an increase in non-minority residents in the area.

However, there are some concerns about the manuscript, which are listed below:

1. The reasons causing less pronounced improvement in air quality over gentrification areas are not solid enough. The authors discussed some very general reasons without any solid evidence. I suggest providing some quantitative evidence (e.g., gas usage and greenspaces changes over time) maybe more persuasive.

2. Between lines 238 and 244, the authors built the GLM model for regression analyses. However, they did not have any discussions and/or explanations about this result. It is confusing what the regression analyses were used in the manuscript, and what are the coefficients of other pollutants.

3. The tables in the manuscript contain too much information, making them difficult to read. It is better if the authors could create some figures to summarize or highlight the most important information, and maybe move the tables into supplementary materials. Time-series plots maybe helpful to illustrate changes over time instead of text and tables.

4. line 62, it is better to emphasize the year of the ‘last United States census tabulation’ in the context directly to avoid any confusion.

Author Response

Thank you for your great comments.  Attached is a response to each reviewer.  Every comment has been answered.  Thank you.

Reviewer 2 Report

This article written based on the premise that gentrification was strongly associated with racial distribution and racial distribution also associated with air quality. Furthermore, gentrification related with air quality as the title. From a commonsense standpoint, we agree that change of racial distribution influence give some effect to urban environment and the change of urban environment could give some effect to the air quality too.

Otherwise, as author mentioned at the abstract and discussion, gentrification is occurring in large cities throughout the world. The most likely reason for gentrification is diverse, such as substantial economic investment by local and national investors into the city. And these investments often involve the demolition of existing buildings and the construction of new ones, often cause environmental pollution as the inevitable reward. In addition to this, a wide variety of socioeconomic factors also influence gentrification. From this general point of view, it must be pointed out that there are some important logical deficiencies in the conceptual structure of this paper.

First, author strongly dependent on the naive premise that since ‘A’ and ‘B’ are related and ‘c’ and ‘B’ are related, ‘A’ and ‘c’ are related.Since gentrification was strongly associated with racial distribution and racial distribution influence air quality, gentrification necessarily have inevitable relationship with air quality. 

But if we recognize there are various causal factors of gentrification including socioeconomic factors, it should be easy to understand that racial distribution is only one of many factors. Therefore, attempts to directly link the causal relationship between gentrification and racial distribution is not logically appropriate. Furthermore, causal relationship between racial distribution and air quality is not logically appropriate, because there many direct causal factors of air quality. 

Second, Characteristics of Gentrification and Median levels of pollutants by Decade were analyzed and compared differences between areas classified by the levels of gentrification during each decade in each zip code of Wayne County in order to demonstrate causal relationship between gentrificationracial distribution and air quality. Nonetheless, no significant differences are seen in the data between areas.

Finally, because consistency between the gentrificationracial distribution and air quality are not recognized from the data, at least analysis needs to be conducted while controlling for other dependent variables if relationship to be demonstrated. 

Author Response

Thank you for your comments. They helped us to enhance the manuscript tremendously. Each comment has been answered. Thank you.

Round 2

Reviewer 1 Report

Thanks for the authors' revision based on the comments. I think the manuscript is in a good format. 

Author Response

Comment: Thanks for the authors' revision based on the comments. I think the manuscript is in a good format. 

Response: Thank you very much for your initial comments and critiques.  They helped us signficantly in revising the manuscript to make it acceptable.  Thanks.

Reviewer 2 Report

Thanks for your revised version. I agree that it is useful for the general public to be aware of these associations between the two phenomena too. Nonetheless, still it should be pointed out that the incorrect in that initial assumption in Abstract(14-16 lines) need to be revised as added in Discussion also.

Author Response

Comment: Thanks for your revised version. I agree that it is useful for the general public to be aware of these associations between the two phenomena too. Nonetheless, still it should be pointed out that the incorrect in that initial assumption in Abstract(14-16 lines) need to be revised as added in Discussion also.

Response: Thank you for this point.  We agree that the abstract should also be revised accordingly to emphasize that causality is not proven at all.  We have changed that part of the abstract now to read "To investigate this association, we studied the trends of gentrification, changes in racial distribution and changes in air quality in each zip code of a large urban county over a 40-year period."